

# First DNA barcode library for the ichthyofauna of the Jos Plateau (Nigeria) with comments on potential undescribed fish species

Michael Olaoluwa Popoola[1], Frédéric Dieter Benedikt Schedel[2,3], Paul DN Hebert[4] and Ulrich Kurt Schliewen[3]

[1] Obafemi Awolowo University, Ile-Ife, Nigeria
[2] University of Basel, Basel, Switzerland
[3] SNSB-Bavarian State Collection Zoology (ZSM), München, Germany
[4] University of Guelph, Guelph, Canada

Corresponding author
Michael Olaoluwa Popoola,
popoolam@oauife.edu.ng

## ABSTRACT

Located in the central region of northern Nigeria, the Jos Plateau covers approximately 9,400 km² with an average altitude of 1,280 m and constitutes a unique terrestrial ecoregion known as the Jos Plateau forest-grassland mosaic. The biota of the Jos Plateau include endemic elements, but very limited information is available on its ichthyofauna. This is despite the fact that the ancient plateau contributes to several large rivers spanning multiple major drainage systems including the Niger and Benue Rivers, and Lake Chad. This study provides the first species list for the fishes of the Jos Plateau based mainly on 175 DNA barcoded museum voucher specimens representing 20 species, and another three species without a DNA barcode. In total, 23 species from eight families and 17 genera were collected from the Jos Plateau including five putatively new species, four in the family Cyprinidae and one in the Clariidae. With ten species, the Cyprinidae is the most diverse fish family on the Jos Plateau, followed by Clariidae and Cichlidae, each with three species. The study also provides data on species distribution and habitat parameters including information on water chemistry that strongly suggests that selected water bodies are heavily impacted by anthropogenic activities. Urgent management steps are required to preserve the unique and diverse fish communities of the Jos Plateau and their habitats.

## INTRODUCTION

Scientific interest in the aquatic biodiversity of the African continent dates back to the 18th century. One important step towards an objective basis for the comparative study of its aquatic biodiversity involved dividing the continent into more or less homogenous fish fauna regions or ichthyological provinces (*Leveque, 1997*; *Roberts, 1975*). The Jos Plateau forest-grassland mosaic falls within the Nilo-Sudan ichthyological province (*Stiassny, Teugel & Hopkins, 2007*) and is well known as a biogeographically distinct terrestrial ecoregion (*Wright & Jones, 2005*). However, the fish fauna of the Jos Plateau has received

very little attention. This is surprising because the rivers that drain from this high plateau (~1,000 m) are unique in that they are mostly isolated from downstream sections by waterfalls, draining geologically old granitic and basalt formations of different origin that span very different watersheds including the Niger River and the Chad and Benue Basins (*Buchanan & Pugh, 1955*).

With a surface area of 9,400 km$^2$, the Jos Plateau is located in the central part of northern Nigeria and lies at an average elevation of 1,280 m above sea level, with geologically ancient granite hills and rock outcrops sometimes reaching another 300 m in elevation (*Lee, 1972*). The area is drained by several major rivers which are important sources of domestic water for this densely populated region. The watershed pattern on the Jos Plateau is unusual as its rivers drain into three major river systems in Nigeria: rivers flowing northeast drain into the Kano River and Lake Chad (Chad drainage); east flowing rivers drain into the Gongola River, a tributary of the Benue River drainage; rivers flowing south drain as well to the Benue River; and west flowing rivers drain into the Kaduna River which feeds the Niger River (Niger drainage system). Seventeen rivers; Delimi, N'gell, Gurum, Rukuba, Jarawa, Shen-fusa, Foron, Kassa, Assop, Daffo, Gindiri, Tahoss, Korot, Magurji, Maijuju, Bokkos and Farin-ruwa (see Table S1) were surveyed in this study. The Delimi River, however, is the main drainage system of the Jos Plateau and a major tributary of the Shari River system which flows north-east before it drains into Lake Chad, covering a distance of about 900 km.

The Jos Plateau has a wet-dry climate which is largely determined by its elevation and position across the seasonal shifts of the Inter-Tropical Convergence Zone (ITCZ). Generally, the average temperatures on the Jos Plateau are lower than those in the rest of Nigeria. The dry season is dominated by the north-east trade winds while the beginning of the rainy season is marked by thunderstorms of high intensity. The plateau lies within the northern Guinea Savanna vegetation zone, an open woodland with tall grasses. Though the area has its own unique vegetation, this has been considerably altered by human activities including mining, agriculture, grazing, and the demand for timber and fire wood (*Olowolafe, 2008*).

In fact, the only fish taxa described from the greater Jos Plateau are the nothobranchiid killifish *Fundulopanchax gardneri nigerianus Clausen, 1963* and the labeonin cyprinid *Garra trewavasae Monod, 1950*. Moreover, extremely few ichthyological records are available in museum collections as reflected in global databases, such as Fishbase (*Froese & Pauly, 2019*). However, *Reid & Sydenham (1979)* surveyed part of the Jos Plateau in their study of the fishes of the Lower Benue. Other studies on the fishes from the Plateau are unpublished theses (*Antony, Eneriene & Ufodike, 1986*). However, these studies either only refer to commercially important fish species, some of which have been introduced (*Oreochromis niloticus Linnaeus, 1758*, *Poecilia reticulata Peters, 1859*, *Cyprinus carpio Linnaeus, 1758*), while there is little information on smaller fish species. Further, these studies were not designed to provide faunistic baseline data and lack up-to-date systematic ichthyological expertise. As recent studies have shown; some water bodies on the Jos Plateau have been degraded, and its undocumented fish species are potentially at risk (*Ademola, 2008*; *Akpan & Anadu, 1991*; *Anadu & Ejike, 1981*).
This study provides first comprehensive information on the ichthyological diversity of the Jos Plateau based on DNA barcodes and morphological identification of voucher specimens.

## MATERIALS AND METHODS

### Fieldwork, study sites and species identification

Sampling was carried out on several field trips to the Jos Plateau during the dry season (December 2013 and January-April 2014). Fishes were collected at 37 strategically selected sampling locations in 17 major rivers on the Jos Plateau (Table S1 provides detailed locality information, Figs. 1 and 2). Fishes were captured using different methods (cast net, gill net, frame net, fish traps) depending on the size and habitat of the site. Freshly caught specimens were euthanized with an overdose of anaesthetic (Benzocaine, MS-222). Specimens representative of all taxa from each collection at each sampling site were individually tagged and pectoral fin clip sampled. The whole specimens were photographed after-which they were fixed in 5% formalin and eventually preserved in 70% ethanol after being treated in changes of water followed by different concentrations (20%, 40% and 60%) of ethanol according to *Neumann (2010)*. Tissues were placed immediately into individually labelled tubes filled with 96% ethanol. Specimens were exported on the 17th of September, 2014 and subsequently deposited in the fish collection of SNSB-ZSM (Germany). We followed all applicable international and national guidelines of animal use and ethical standards for the collection of samples. The permit to survey the fishes of the Jos area was obtained from the GWOM SOPP palace, Riyom, Plateau State. After a preliminary identification of specimens in the field to a genus level, preserved specimens were subsequently assigned to a species using standard identification keys (*Paugy, Leveque & Teugels, 2003*; *Stiassny, Teugel & Hopkins, 2007*). When required, additional literature (*i.e.* original descriptions) were consulted allowing species identification and the clear identification of diagnostic characteristics. This was especially necessary for the identification of the candidate species (following the terminology regarding potentially undescribed species of (*Padial et al., 2010*; *Vieites et al., 2009*)) belonging to the genera *Enteromius* and *Clarias* and their comparison to potentially similar species. When morphometric measurements were required, we measured those following protocols in the identification keys using a manual calliper. Important meristic traits were examined using a stereomicroscope (Leica MZ6) as well as with a digital X-ray using a Faxitron Ultrafocus LLC X-ray unit. However, some of the collected specimens could not be assigned to any described species known from West Africa (please see further below for a comprehensive overview of the newly discovered candidate species).

### Notes on collection sites (see Fig. 2)

Most of the collection sites were situated in perennial or annual headwaters of major rivers in Nigeria. Generally, they originate as small streams before they flow over long stretches of fine-grained alluvial soils with few rocks. After a substantial increase in stream size, they reach the margin of the high plateau before descending, often in rapids and cascades, to the lowland plains below. On the plateau, water flow is usually moderate, whereas in

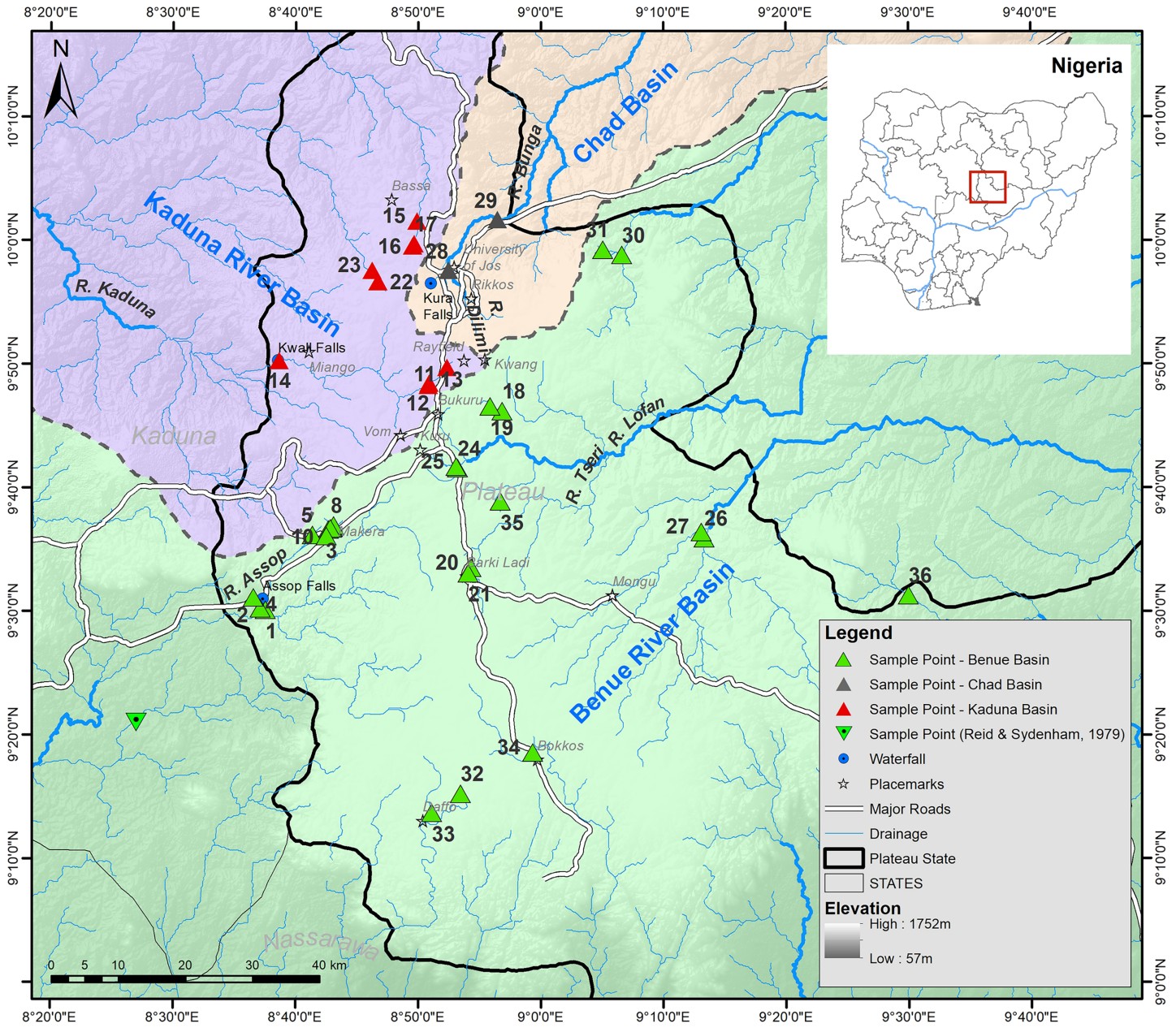

**Figure 1** Map of the Jos Plateau area showing the sampling locations (green, red and grey triangles) examined in this study and the sampling point (green inverted triangle) examined by *Reid & Sydenham (1979)*.

rapids and waterfalls it may be torrential. Vegetation along the river banks provides partial shading and is best characterized as a savanna-forest mosaic. None of the sampled locations was pristine; all were impacted by farming, domestic waste deposition, industrial discharges or by mining.

### Water chemistry (Table S3)

The physico-chemical characteristics of the Assop, N'gell, Rukuba, and Tahoss rivers were assessed following standard procedures. The water temperature was measured *in situ* using

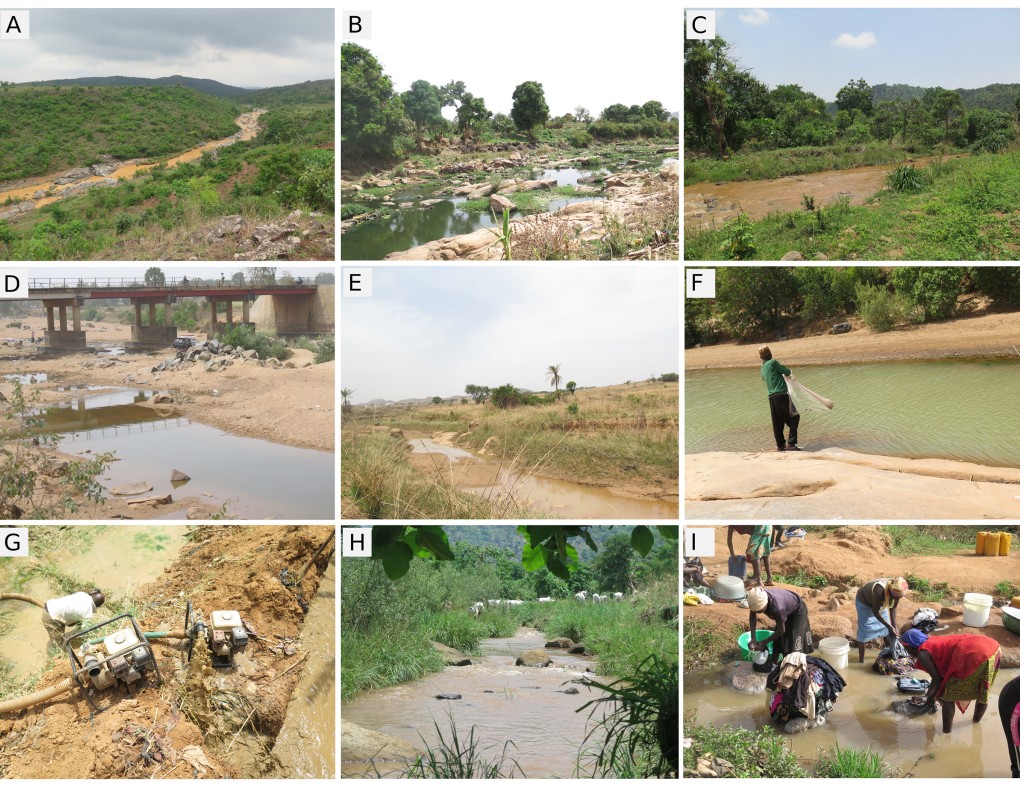

**Figure 2 Overview of sampling sites: Table S1 (A) Magurji; (B) Delimi; (C) Assop; (D) Gindiri; (E) Shen Fusa; (F) Fishing, Magurji; (G) Artisanal mining, N'gell River; (H) Cattle trough, Assop; (I) Washing, Tahoss.** Photo credit: Popoola Michael Olaoluwa.

mercury-in-glass bulb thermometer with calibration range (−10 °C to 110 °C). Samples for the analysis of physico-chemical parameters were collected in clean 2 litre plastic bottles. However, dark reagent bottles were used for Biochemical Oxygen Demand ($BOD_5$) sample collections and kept in a dark cupboard for 5 days for subsequent analysis. The pH, conductivity, and total dissolved solids (TDS) of each sample were determined using a PCE-PHD1 multi-parameter meter. The analytical determinations of the physico-chemical water quality parameters were carried out within the holding time of each parameter (*Ademoroti, 1996*; *APHA, 1995*; *Golterman, Clymo & Ohnstadt, 1978*). The $BOD_5$ was determined by iodiometric titration (*Golterman, Clymo & Ohnstadt, 1978*). Colour was investigated using a colourimeter (Jenway 6051 model) standardized with a set of potassium chloroplatinate-cobalt solutions as Pt-Co standards, while turbidity was assessed using a turbidimeter (*Ademoroti, 1996*). Sulphate ($SO_4^{2-}$) and nitrate ($NO_3^-$) were determined by spectrophotometric methods (*Ademoroti, 1996*). Magnesium ($Mg^{2+}$) was determined by the complexio-metric titration method using $Na_2$EDTA (*Golterman, Clymo & Ohnstadt, 1978*). All recommended quality control (QC) and quality assurance (QA) measures were taken for each determination.

## Laboratory protocols

For DNA extraction, we subsampled the ethanol-preserved fin clips and transferred them to 96-well plates. These plates were sent to the Canadian Center for DNA Barcoding (CCDB, Guelph, Canada) for DNA extraction, PCR amplification, and Sanger sequencing (all protocols are available online: ccdb.ca/resources/). PCR amplification of all samples employed the fish primer pair C_FishF1t1/C_FishR1t1 (*Ivanova et al., 2007*) for the CO1 (mitochondrial cytochrome *c* oxidase subunit 1) barcode fragment. The same primers were used for subsequent bidirectional Sanger sequencing reactions. The resulting sequence data and trace files were uploaded to barcode of life database (BOLD) (*Ratnasingham & Hebert, 2007*) as well as all corresponding metadata voucher information (*i.e.*, locality data, altitude, taxonomic classification, habitat images, primer information etc.) and are publicly accessible on BOLD in dataset DS-NGAJO (DOI 10.5883/DS-NGAJO).

## Molecular data anaylsis

We used the "Sequence Analysis" toolbox on BOLD and restricted analysis to sequences >500 bp. MUSCLE (*Edgar, 2004*) was employed to align the sequences while the sequence divergences (*i.e.*, mean and maximum intraspecific variation as well as the minimum genetic distance to the nearest-neighbour species) were calculated using the 'Barcode Gap Analysis' tool on BOLD. This analysis employs the Kimura-2-Parameter (K2P) distance metric (*Puillandre et al., 2012*). As well a neighbour-joining (NJ) tree was constructed to graphically represent the BIN clusters. Within BOLD, closely similar COI barcode sequences are assigned a globally unique identifier, termed a "Barcode Index Number" or BIN (*Ratnasingham & Hebert, 2013*). The "BIN Discordance" report on BOLD was used to reveal cases where BINs were shared between species clusters or restricted to certain species clusters (BIN Discordance and BIN Sharing). In addition, we obtained for each BIN recovered on the Jos Plateau the nearest BIN available on BOLD (which includes data deposited on GenBank) along with corresponding divergence information directly from the individual BIN records (see Table S4). This was done by screening the BOLD database for all available CO1 sequences for each genus (with a minimum length of 500 bp) using the "Record Search" tool, including public records mined from GenBank.

In addition, we created additional and more comprehensive CO1 DNA barcode sequence datasets for all genera for which putatively undescribed species (see below) appeared to be present on the Jos Plateau region. This was equally done by screening the BOLD database for all available CO1 sequences for each genus (with a minimum length of 500 bp) using the "Record Search" tool. Subsequently, we constructed a NJ tree for each genus as indicated above.

## RESULTS

Nearly 800 specimens were collected during the two surveys and, based on morphological keys, they were assigned to 23 species (eight families, 17 genera), some of which appeared to represent undescribed species (see below for further information). Potentially new species fell into four genera-*Clarias Scopoli, 1777*, *Enteromius Cope, 1867*, *Labeo Cuvier, 1816* and *Labeobarbus Rüppell, 1835*. Further, two species were observed to have different

phenotypes (morphs)-*Coptodon zillii* (*Gervais, 1848*) and *Enteromius perince* (*Rüppell, 1835*). *Poecilia reticulata Peters, 1859* was the only non-native species found on the Jos Plateau. The order and family with the highest species diversity was the Cypriniformes (family Cyprinidae) with 10 of 23 species so it represented nearly half of all recorded species. It was followed by the Siluriformes (Mochokidae, Clariidae) with five species and by Cichliformes (Cichlidae) with three species. The Osteoglossiformes (Mormyridae) and Cyprinodontiformes (Nothobranchiidae, Poecilidae) were each represented by two species while there was a single species of Characiformes (Alestidae). Representative specimens of all species and morphs found on the Jos Plateau are depicted in Figs. S1–S4. The distribution of the species according to sampling sites are noted in Table S2.

## DNA barcode library

In total, 176 CO1 DNA barcode sequences were generated with lengths >635 bp (1 shorter barcode-444 bp, was excluded from further sequence analysis). These sequences represented 20 of the 23 fish species collected on the Jos Plateau. Three species lacked barcode coverage because of suboptimal tissue preservation and low sample size: *Chiloglanis* cf. *benuensis Daget & Stauch, 1963*, *Clarias* cf. *gariepinus Burchell, 1822* and *Marcusenius mento Boulenger, 1890*. Only a single barcode was obtained for five species (*Heterobranchus longifillis*, *Labeo sp.* Assop, *Labeobarbus bynni*, *Synodontis violaceus*, and *Poecilia reticulata*), whereas the other species all had two or more sequences. Overall, CO1 amplification was successful for 92% of the 190 submitted specimens. The sequence divergence values for the CO1 barcode region (Table S4) show that most species of the Jos Plateau possessed very low intraspecific genetic distance values.

An NJ tree (Fig. 3) showed that barcodes reliably discriminated most species as all conspecific specimens were recovered as monophyletic lineages assigned to a single BIN (19 BINs in total). *Oreochromis niloticus* and *Sarotherodon galilaeus* shared a BIN, but formed two different haplotype clusters with a low sequence divergence (see Fig. 3). These results are similar to those in an earlier barcoding study on the freshwater fishes of south-eastern Nigeria (*Nwani et al., 2011*) which found very low genetic divergence between *O. niloticus* and *Sarotherodon galilaeus boulengeri*. While another barcoding study on the ichthyological fauna of the Nile recovered a low genetic divergence between *Oreochromis aureus* (*Steindachner, 1864*) and *S. galilaeus* (*Ali, Ismail & Aly, 2020*). The clustering of *S. galilaeus* within *Oreochromis* might either point to regular mis-identification of this species, which however appears rather unlikely, or a taxonomic misassignment of *S. galilaeus* in the genus *Oreochromis*. However, recent nuclear based studies support the monophyly of the genus *Oreochromis* while *Sarotherodon* clearly appears to be polyphyletic (*Dunz & Schliewen, 2013*; *Ford et al., 2019*). Overall, the afore-mentioned studies in combination with our results alternatively suggest a cytonuclear discordance in respect of the phylogenetic position of *S. galilaeus*, which might be the result of past hybridization events. Indeed, hybridization between *Oreochromis* and *Sarotherodon* has been previously suggested to have taken place (see *Ford et al., 2019*; *Nagl et al., 2001*). Further, it is known that both genera hybridize in captivity (*Otubusin, 1988*) and are commonly farmed in aquaculture facilities throughout Nigeria
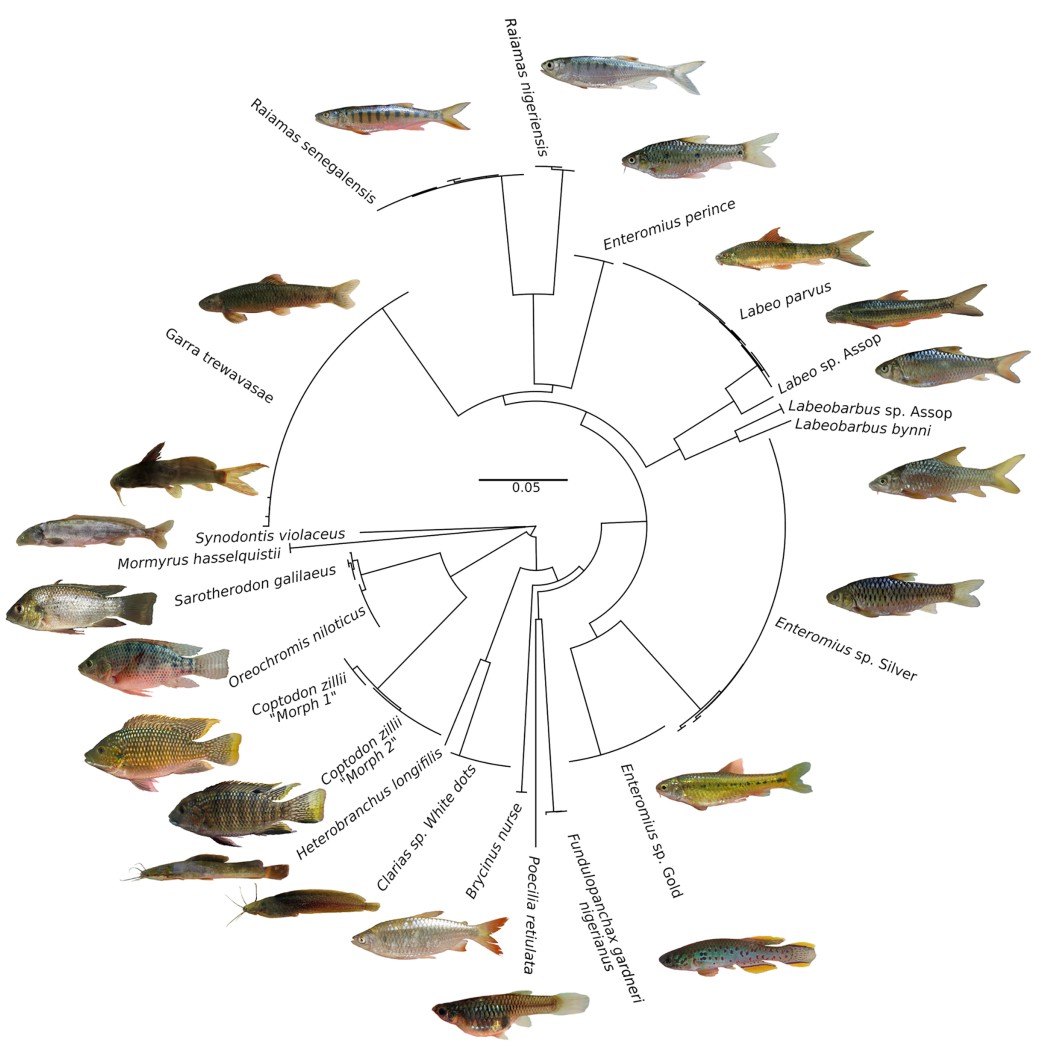

**Figure 3 Neighbour-Joining tree based on 176 CO1 barcode sequences of fish species occurring in the Jos Plateau, created in BOLD using "Taxon ID tree" tool.** The circularized NJ tree was created using FigTree v1.4.4 (https://github.com/rambaut/figtree/releases). Photo credit: Popoola Michael Olaoluwa.

(*Ayinla, 2007*). In any case, the biological reason for the BIN sharing between the *O. niloticus* and *S. galilaeus* on the Jos Plateau and elsewhere remain uncertain and would need to be critically tested with appropriate genomic datasets.

## Notes on new candidate species from the Jos Plateau

The morphological investigation of the collected specimens as well as the subsequent COI analyses suggest the presence of at least five candidate species, four Cyprinidae, and one Clariidae. Among the potentially new cyprinid species, two belong to the genus *Enteromius* (*Cope, 1867*). This genus was recently revalidated by (*Yang et al., 2015*) for the small diploid African barbs formerly placed in the genus *Barbus* (*Daudin, 1805*), but is known to be polyphyletic (*Yang et al., 2015*). Although a comprehensive taxonomic re-evaluation of the diploid African barbs is needed, this interim generic assignment is

currently generally accepted (*Englmaier, Tesfaye & Bogutskaya, 2020*; *Hayes & Armbruster, 2017*; *Martin & Chakona, 2019*; *Skelton, 2016*).

The first species, herein referred to as *Enteromius* sp. Silver (see Fig. S2), was widespread on the Jos Plateau, being present at most collection sites. It is phenotypically similar to *Enteromius callipterus* (*Boulenger, 1907*), a widely distributed West African barb originally described from Cameroon but differs by its possession of a faint greyish longitudinal band (*vs* no lateral colour pattern for *E. callipterus*). In addition, the dorsal spot in the dorsal fin is less pronounced, *i.e.*, irregular and not deep black but rather greyish with translucent orange (background colour of the dorsal fin), as compared with *E. callipterus* (*Leveque, 2003*). In a NJ tree based on all available CO1 sequences of *Enteromius* on BOLD (see Fig. S5) *Enteromius* sp. Silver represents a distinct mitochondrial lineage not clustering with *E. callipterus*. Instead, it was sister group to a cluster including different *Enteromius* species occurring in the Congo drainage *i.e., Enteromius brazzai* (*Pellegrin, 1901*) and the Zambezi drainage *i.e., Enteromius radiatus* (*Peters, 1854*).

The second candidate species, referred herein as *Enteromius* sp. Gold (see Fig. S2) was collected from four rivers (N'gell, Delimi, Shen Fusa, Rukuba). It is a comparatively small species easily recognized by its yellowish to golden body coloration and small irregular blackish spots forming a more or less confluent longitudinal band on the sides which terminates before the base of the caudal fin in a slightly larger blackish spot. Morphologically, *Enteromius* sp. Gold could not be assigned to any *Enteromius* species known from West Africa or Lower Guinea based on standard keys (*Leveque, 2003*; *de Weirdt et al., 2007*). Interestingly, a NJ tree based on all CO1 sequences for *Enteromius* on BOLD (see Fig. S5) revealed that *Enteromius* sp. Gold was a sister taxon to *Enteromius* cf. *macrotaenia* (*Worthington, 1933*), a species known from south-eastern Africa. The sister group of this cluster included *Enteromius lukusiensis* (*David & Poll, 1937*), *Enteromius greenwoodi* (*Poll, 1967*), and the goldie barb *Enteromius pallidus* (*Smith, 1841*), which is endemic to the eastern Cape Fold Ecoregion of South Africa (*Martin & Chakona, 2019*). These results suggest the closest relatives of *Enteromius* sp. Gold are found in the southern parts of Africa, raising interesting questions on its biogeographic history (*i.e.*, when and how it colonized the Jos Plateau area).

The third potentially new cyprinid species, referred herein as *Labeo* sp. Assop, was only collected from the Assop River (see Fig. S2). It shows affinities with *Labeo parvus Boulenger, 1902* which is widespread on the Jos Plateau. A NJ tree including all available CO1 sequences for *Labeo* on BOLD (see Fig. S6) revealed that *Labeo* sp. Assop was a sister taxon to a clade including *Labeo lukululae Boulenger, 1902 Labeo quadribarbis Poll & Gosse, 1963*, and *Labeo* sp. aff. *rectipinnis Tshibwabwa, 1997*. Interestingly, and similar to the case of *Enteromius* sp. Gold, all three species forming the sister group to *Labeo* sp. Assop are neither present in the Niger/Benue drainage system nor in the Chad system but are restricted to the Congo drainage system.

The fourth potentially new cyprinid species, referred herein as *Labeobarbus* sp. Assop, is the second candidate species collected only from the Assop River (see Fig. S3A). It could not be assigned to any known *Labeobarbus* species found in western Africa based on morphological data and following the key in *Leveque (2003)*. The collected specimens

were recovered as sister to *Labeobarbus brevispinis* (*Holly, 1927*), a species known from Lower Guinea but as well from the Benue River drainage (*de Weirdt et al., 2007*), in a NJ tree based on all available CO1 sequences on BOLD (see Fig. S6).

The last candidate species, referred herein to as *Clarias* sp. White dots (see Fig. S4), was found in four rivers (N'gell, Gurum, Kassa, Rukuba). It could not unambiguously be assigned to any described *Clarias* species known from Western Africa based on standard morphological keys (*Teugels, 2003*; *Teugels et al., 2007*). Nevertheless, it is phenotypically similar to *Clarias agboyienesis Sydenham, 1980*, a species known from West African coastal basins including the lower course of the Niger River. The latter species is, however, characterized by a very light coloration with yellowish brown flanks and a light grey belly whereas the body ground coloration of *Clarias sp*. White dots is dark with larger specimens having small whitish to yellowish spots on the flanks and on unpaired fins. Interestingly, the sequenced specimens of *Clarias sp*. White dots clustered with sequences of *Clarias gabonensis Günther, 1867* in the NJ tree based on all CO1 sequences for *Clarias* on BOLD (see Fig. S7). The coloration of *C. gabonensis* is either uniform brownish yellow or marbled with small yellowish spots (*Hanssens, 2007*; *Teugels et al., 2007*) which shows some correspondence to the coloration pattern of *Clarias* sp. White dots, but it has a distinctively shorter head. Furthermore, *C. gabonensis* occurs in the Congo basin and different drainage systems of Lower Guinea, *e.g.*, the Ogowe, Noya, Kouilou and Chiloango rivers (*Hanssens, 2007*; *Teugels et al., 2007*) and it was only recently recorded from Nigeria (*Nwani et al., 2011*). However, the taxonomic assignment of the Nigerian specimens of *Clarias gabonensis* might be incorrect and it should be carefully investigated, since the identification of *Clarias* species is often challenging. Indeed, the available sequences of *C. gabonensis* were dispersed in distinct clusters in our NJ tree (see Fig. S7), indicating probable misidentifications or the presence of cryptic species. Our sequences of *Clarias* sp. White dots group with some Nigerian *C. gabonensis* in one of the larger clades identified as *C. gabonensis*. Taken together it would be premature to assign our specimens of *Clarias* sp. White dots to *Clarias gabonensis* until more thorough morphological and genetic investigations are made within this species complex.

A thorough taxonomic investigation and the formal description of these candidate species is in progress and will be presented in a follow up study.

In addition to these five putatively new species, we observed substantial phenotypic variation in two species. First, in the cichlid, *Coptodon zillii*, which is widespread on the Jos Plateau and occurs in two color morphs, morph 1 and morph 2. Both morphs occurred sympatrically at one single location (Delimi River). Although differences between these morphs is modest, the flank coloration of one morph is predominantly yellowish (Morph 1) whereas in the other morph is whitish (Morph 1) (see Fig. S1). Furthermore, morph 1 has a pale green upper lip and and a whitish lower lip while morph 2 has greenish to brownish upper lips and blueish lower lips. Members of the two morphs formed two distinct mitochondrial lineages with low genetic divergence (see Fig. 3).

Specimens of *Enteromius perince* (*Rüppell, 1835*) also showed clear intraspecific variation. Most specimens showed the colouration typical of this species (*i.e.*, silver body ground coloration with three spots aligned along the mid-lateral line), but a few specimens

had additional spots. These individuals resemble the so-called "*lepidus* form" (*Leveque, 2003*), but no CO1 barcodes were recovered from them so their divergence from the nominate form is uncertain.

## DISCUSSION

By providing the first DNA barcode library for the ichthyofauna of the Jos Plateau, this study represents an important progress in the taxonomic study of Nigerian fishes as few earlier investigations have employed DNA barcoding (*Falade, Opene & Benson, 2016*; *Iyiola et al., 2018*; *Nneji et al., 2020*; *Nwakanma, Ude & Unachukwu, 2015*; *Nwani et al., 2011*; *Sogbesan et al., 2017*; *Ude et al., 2020*). Aside from providing barcode records for known species, the results suggest the presence of at least five species new to science. As well, the new DNA barcode records constitute an important contribution to the growing reference library for Nigerian freshwater fishes, especially because most of the species are shared with the Niger, Benue, and Chad River drainages, the major river systems in Nigeria.

Most rivers on the Jos Plateau flow into the lower Benue River drainage and hence a part of the wider Nilo-Sudan Ichthylogical Province (*Roberts, 1975*). Among the site of fish collections by *Reid & Sydenham (1979)* in their study of fishes from the lower Benue river basin, two sites were on the Jos Plateau. The authors reported eight species from these two sites. The species reported were *B. nigeriensis* (now known as *E. nigeriensis*), *B. spurelli* (now known as *E. ablabes*), *B* cf. *holasi* (now known as *Labeobarbus* cf. *wurtzi*), *Fundulopanchax gardneri* and a species the authors provisionally identified as *G. waterloti*. The authors also reported two other *Garra* species which they were not certain about their taxonomic identity. One of the *Garra* species was reportedly thought to be either *Garra waterloti* (*Pellegrin, 1935*) or *G. ornata* (*Griscom & Nichols, 1917*), while the other was supposed to be either *G. trewavasae* or *G. waterloti*. However, they could not identify the original collection site for *G. trewavaseae* described by (*Monod, 1950*) as 'a small tributary of the Gongola River, South West of the Bargesh district'.

Just two of the species (*G. trewavasae*, *F. gardneri*) reported by *Reid & Sydenham (1979)* were found during our study, perhaps because we did not resample their sites (see Fig. 1). Therefore, the species count for the Jos Plateau area based on this study might be expanded to include four more species (*E. nigeriensis*, *E. ablabes*, *Labeobarbus* cf. *wurtzi*, *G. waterloti*). Alternatively, some species reported earlier may reflect misidentifications. This might be the case for specimens assigned to "*Barbus spurelli*", a species described by *Boulenger (1913)* from Ghana that was later synonymized with *Barbus ablabes* (now *E. ablabes*) by *Leveque (1983)* as he was unable to find any differences between these two taxa apart from differences in the coloration; these are the presence of a longitudinal stripe on the middle of the sides in *E. ablabes*, which should be absent in "*Barbus spurelli*". *Enteromius* sp. Silver has only a very faint longitudinal stripe and matches with some aspects of the description of "*Barbus spurelli*"; it has, *e.g.*, dark-edged scales which are darker at the base, indicating that our specimens of *E.* sp. Silver might be conspecific with the specimens identified as "*Barbus spurelli*" by *Reid & Sydenham (1979)*. In fact, *E.* sp. Silver is readily distinguished from *E. ablabes* by its possession of longer barbels and more

than 3.5 scales between the origin of the dorsal fin and the lateral line (*Paugy, Leveque & Teugels, 2003*). However, without careful re-examination of fish specimens collected by *Reid & Sydenham (1979)* this remains speculative. It is clear that the sampling locations visited by *Reid & Sydenham (1979)* should be resampled to help resolve the identity of the species at these two localities.

The overall number of fish species (23) recorded from the Jos Plateau is low in comparison with other river basins in Nigeria such as the Imo (142 species) and Ogun (119 species). As well, more than half (12) of the fish species on the Jos Plateau are also found in the Benue, Chad, and Niger drainages (*F. gardneri, M. hasselquisti, B. nurse, E. perince, L. parvus, R. nigeriensis, R. senegalensis, H. longifillis, S. violaceous, O. niloticus, S. galilaeus, T. zillii*). The only described fish species endemic to the area is *Garra trewavasae* and the subspecies *Fundulopanchax gardneri nigerianus* rendering a low regional endemism, and *M. mento* is found in both the Upper Niger and Lower Benue basin but not in the Chad basin. However, if the potentially new species detected in this study are considered, the level of endemicity would reach ~ 30% (7 out 23 recorded species), a result emphasizing the need for further taxonomic and biogeographic research.

The closest relative of *Enteromius* sp. Gold seems to be found in Southern Africa which underscores the important gap in knowledge of the biogeography of the small African *Enteromius*. This lack in knowledge is largely due to the limited understanding of phylogenetic interrelationships within the group despite past studies (*Howes, 1991*; *Rainboth, 1991*). However, the recent study of *Yang et al. (2015)* has shed more light on the biogeographical distribution of the cyprinine fishes. Their study suggests that the distribution of the cyprinids could not have been influenced by Gondwanaland separation as the dispersal events of the cyprinids are much more recent than the separation between 50–80 mya. Their findings suggest several independent dispersal events for cyprinids. Among the Smiliogastrini (diploid barbs), two hypothetical dispersal events were identified; those involving the small-sized African barbs and allies and another involving the small-sized Asian *Puntius* and allies. However, there is inadequate information to explain the factors that might have shaped the odd distribution of the newly found *Enteromius* sp. Gold between western and southern Africa, if the preliminary DNA barcode based association is maintained.

The Jos Plateau is dotted with waterfalls along its main rivers (Farin Ruwa, Kwall, Assop, Kura) at the southern, western, and eastern margins respectively. Waterfalls are ordinarily viewed as a major dispersal barrier to fishes (*Torrente-Vilara et al., 2011*), but the impact of those on the Jos Plateau appear to be limited since upstream and downstream populations of fishes did not show sequence divergence at CO1. Furthermore, not all rivers that originate on the Jos plateau feature a waterfall because the north-eastern part of the Plateau slopes gradually towards the lowlands (*Payne & Firefinch, 1998*). *Garra trewavasae* has apparently not dispersed beyond the Jos Plateau area because of the lack of comparatively cold streams in the lowlands.

The sampling program for this study occurred during the dry season when most water bodies were nearly dry, being fragmented into remnant pools along the river course. Only a few were flowing. This situation suggests that rivers on the Jos Plateau are already heavily

impacted by climate change that has reduced rainfall in the area by more than 5% (*Fasona & Omojola, 2005*). On the positive side, most water quality parameters were within ranges tolerated by fishes except the high turbidity and colour of many water bodies, especially on the N'gell River (Table S3). The fact that all assessed water bodies were turbid underscores widespread anthropogenic impacts across the plateau, indicating the need for urgent management steps to safeguard fish communities from intensified impacts in the future. Some of the key activities leading to this high turbidity are domestic washing, small-scale mining, making of building blocks, and the release of industrial waste (Fig. 2).

## CONCLUSION

In conclusion, this study provides a valuable COI barcode dataset which will serve as reference for further studies on the ichthylogical diversity of western Africa. It indicates the need for further taxonomic studies on Nigerian fishes with an obvious need to evaluate the taxonomic status of the five potentially new species found on the Jos Plateau.

## ACKNOWLEDGEMENTS

We thank Dr. Tinuke Olaleye and Dr. Pam Luka, who worked hand-in-hand to provide logistics support, information about the local geography and the security situation which were invaluable for the success of our field work. Mr Samuel Popoola helped to design the map of the Jos Plateau. Emmanuel Ikpe collected the water samples while he and Dr. Adedeji Aduwo helped with the water quality analysis. We thank Lawrence Kent for his kind support and encouragement to build an ichthyological collection at Obafemi Awolowo University. Our appreciation also goes to Jérôme Morinière (AIM, Advanced Identification methods, Leipzig) for his support and useful advice for handling the BOLD database. We are grateful for the kind support of the CBG team (Guelph, Canada) in processing the samples and providing results on the BOLD database. Dirk Neumann (SSNB-ZSM) is kindly acknowledged for his help with collection management, accessioning, and cataloging the Jos plateau fishes at SSNB-ZSM.

### Funding

Financial support for this study was provided by the International Foundation for Science (IFS A/5361-1 (2013-2016)) while sequencing of about 300 DNA samples were sponsored by the Centre for Biodiversity Genomics, University of Guelph, Canada. Additional financial support was provided by TETFUND Nigeria for the second phase of field work. The funders had no role in study design, data collection and analysis, decision to publish, or preparation of the manuscript.

### Grant Disclosures

The following grant information was disclosed by the authors:
International Foundation for Science: IFS A/5361-1.

Centre for Biodiversity Genomics, University of Guelph, Canada.
TETFUND Nigeria for the Second Phase of Field Work.

## Competing Interests

The authors declare that they have no competing interests.

## Author Contributions

- Michael Olaoluwa Popoola conceived and designed the experiments, performed the experiments, analyzed the data, prepared figures and/or tables, authored or reviewed drafts of the paper, and approved the final draft.
- Frédéric Dieter Benedikt Schedel performed the experiments, analyzed the data, prepared figures and/or tables, authored or reviewed drafts of the paper, and approved the final draft.
- Paul DN Hebert analyzed the data, authored or reviewed drafts of the paper, and approved the final draft.
- Ulrich Kurt Schliewen conceived and designed the experiments, performed the experiments, authored or reviewed drafts of the paper, and approved the final draft.

## Field Study Permissions

The following information was supplied relating to field study approvals (*i.e.*, approving body and any reference numbers):

The permit to collect fishes at the area was provided by the GWOM SOPP PALACE.

## Data Availability

The sequences generated in this study are available at BOLD Systems: DS-NGAJO (DOI 10.5883/DS-NGAJO).

The preserved specimens are deposited in the Bavarian State Collections of Icthyology.

## Supplemental Information

Supplemental information for this article can be found online at http://dx.doi.org/10.7717/peerj.13049#supplemental-information.

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
