# Peer review of "First DNA barcode library for the ichthyofauna of the Jos Plateau (Nigeria) with comments on potential undescribed fish species"

_PeerJ, doi:10.7717/peerj.13049_

## Round 0.1 · original submission · Major Revisions

Dear Authors,

I received three review of your manuscript. All the reviewers were positive, but there are a number of issues that need to be addressed.

1) Organization. I agree that it would be important to introduce the Jos Plateau and its geological/biogeographical context, and then move onto the fish fauna. This should be accompanied by a map that clearly shows major rivers and drainage basins. In the current map, the rivers are difficult to see, and so either change the color of the river, or use a different color scheme to represent elevational range.

2) It would be important to include information on the map from where these samples were obtained. Do they represent different drainage basins? If so are species shared between river basins, or are they restricted to river basins? Understanding of the geographic context of the distribution of the diversity is important for the understanding of broader biodiversity patterns.

3) In principle low divergence between Oreochromis niloticus and Sarotherodon galilaeus boulengeri cannot be due to hybridization. There still is 0.46% sequence divergence. Was there haplotype sharing? Potentially one or both species can be hybrids, but it would have to be with different populations of these species, and the species would need to be geographically structured.

4) At the moment you are only analyzing the specimens you collected. This makes it difficult to conclude anything about interspecific divergences, the existence of new species or potential cases of hybridizations and introductions. You would need to include sequence information from other African fish species, this means using sequences from Genbank in addition to BOLD. While I do not doubt that there are many still to be discovered species, the conclusion that certain specimens are likely new species entirely depends on the information available in the databases to which these specimens are being compared to. So either include these additional data, or modify your interpretations/conclusions accordingly.

5) Finally, there are also orthographic mistakes that need to be fixed.

Otherwise, a nice contribution that hopefully will serve as an impetus for additional molecular biodiversity studies. I look forward to receiving your revision shortly.

Sincerely,

Tomas Hrbek

·

Basic reporting

After carefully reading the manuscript, I found many small typographical errors and few sentences that were not clear and of difficult comphrehension. I added in the revised PDF, comments and suggestions on how o improve it.

The structure of the Introduction can be improved. I suggested some paragraph reorganization, making a a first block directed to the description of the Jos Plateau and a second block about its fish diversity. Previously, some parts of the text focused on the fish then it was followed by an abrupt change to the physical description of the Plateau and going back to the fish and to the Plateau again. My suggestions (on the PDF) is only one idea on how the authors can improve that section.

Two Supplementary tables contained minor typographical erros, and I am also attaching to this review (Supplementary files 9 and 11, table 2and 4, respectively). Also, I could not find any caption for the supplementary figures 2, 3 and 4, and I also noted that for some of the specimens (differentiated by letters) the scale bar is missing. These are: Supplementary figure 2, letter b; supplementary figure 3, letter e; supplementary figure 4, letter f.

All the genera authorships are between parenthesis, but this should not be the case. Please revise all the manuscript.

The Map figure need to be improved. I could barely see the river names (in grey), and also I could not have any idea of which rivers flow into each of the three main watersheds in the region, the Benue, the Niger and the Chad. In a next version of the map, I expect to see at least a small map, showing the three main watersheds, where the Jos plateau is located, and the watershed divides (maybe hashed lines).

Experimental design

no comment

Validity of the findings

no comment

Additional comments

Dear Authors,

It was really nice to see that there are ongoing research related to the fish diversity in the Jos Plateau, a region with not many species, but as suggested by the results, a really heterogenic fauna, sharing close affinities with other African ichthyoprovinces. The article is straigthforward and the conclusions take into account the limitation due to the use of only one marker (COI). It is a great contribution for future fish surveys in west Africa and a nice model to be followed by other similar research in Africa.

As I mentioned in the Basic reporting section, there are many small issues, related to the writting, some corrections that need to be done in the supplementary files and the main issue in my point of view is the map. The map is not clear and need many improvements.

Given the commens I made, I consider that the paper should go through minor revision.

Sincerely,

Pedro Bragança

Reviewer 2 ·

Basic reporting

no commentThe

Experimental design

no comment

Validity of the findings

The study has impact and novelty. All data is provided and is technically sound. Also the conclusions are well stated. This work represent a great contribution to discover an unknown biodiversity in a few studied area. Only, I would have liked to read more discussion about Coptodon morphs related only in results.

Reviewer 3 ·

Basic reporting

The paper adresses the ichtyofauna of an understudied region in Nigeria. It provides an overview of this fauna with molecular data, and voucher specimens. The data is available publicly in BOLD and contains the required info and additional useful data. Export and collect permit were provided although I cannot judge whether they completely fulfill the legal requirements. The paper is generally clear and well written, although there are a number of mistakes to correct (image references and others) and clarifications to be provided, especially in the figures. The number of species is low, but the number of specimens is appropriate and in line with other barcode studies.
The discussion of the new potential species needs to be clarified and criticized on the characters (coloration for Clarias, etc.)

- The map is crucial for this publication, yet it lacks some of the relevant information. The color contrast between rivers and surrounding areas is not very legible, and should be ameliorated.
Ichtyological diversity is often structured according to basins, and the rivers from the plateau drain in several different basins. A clear distinction on the sampling map of the different basins (for instance with different color dots), as well as waterfalls and other obstacles to fish circulation would be needed. Also add the numbers from table 8 on the points (most are clearly distinct, the rest could be indicated with arrows). All these would help strongly with the discussion on waterfalls and sampling points.
- Please add the corresponding collection point numbers on fig 2 for each picture
- L209 if there was hybridisation, you would not observe distinct clusters. Hybridisation cannot make mitochondrial sequences closer to each other, so another hypothesis is needed
The supplementary figures are numbered differently in the text and the files (5, 6, 7 in the files and 2D, G… in the text), and the one for Labeo is missing

Identification: what were the complicated cases ? Please list them

Table 4 the 3 last columns are not very interesting, as they indicate the closest species in the dataset. This distance and species are really not representative of the interspecific divergence of the species with their closest neighbour (absent from the datasets). This should be replaced by the closest species in BOLD and divergence values from them.

Experimental design

Sampling level of completeness should be discussed, considering the low number of species.

Validity of the findings

See above

Additional comments

L45 Fundulopanchax gardneri nigerianus is widely present in the aquarium trade.
L56 , not ;
L66 tributary
rivers flowing south drain into the Benue River as well
L91 supplementary
part starting L113 please add the station numbers corresponding to the various categories
L116 on average ?
L156 Analysis
L161 Puillandre et al., 2012 is not the right reference for K2P on BOLD, or add following but they were not the first.
L200 the specimens you collected have low divergence, but the trees in additional data have higher divergence when including specimens from other locations.
L203 add the BIN numbers on the figure ?
L209 if there was hybridisation, you would not observe distinct clusters ? Hybridisation cannot make mitochondrial sequences closer to each other
L231 divergence with E. callipterus, and closest species in BOLD ?
L242 this is a NJ tree so cluster, not clade, and not optimal for relationships. For phylogenetic relationship discussions, maximum likelihood would be the right approach. For divergence NJ is fine
L251-253 where are the areas of collection for these species ? Same for the other closest species when not indicated (L257), it would be really valuable to have the info.
L341 how do you explain the low species number ?


Supplementary is written 3 different ways in the file titles.
Additional figure 7: Kalign not Kaling
Additional trees: it would be interesting to add indicate the corresponding localities next to the tree, to make them standalone,

---

## Round 0.2 · Minor Revisions

Dear Authors,
I have received two reviews of your revised manuscript. Both reviewers were happy with how you addressed the previous round of reviews, but there are still minor points that need to be addressed.

I would appreciate it if you take this opportunity to address these points and make relevant changes to your manuscript. Once that is done, I will be happy to accept the manuscript.

All the best,
Tomas Hrbek

·

Basic reporting

Dear Authors,

After carefully reading the new version and reading the reply letter I am happy to indicate the paper to a minor revision section, not depending on a third review round.

I found only minor problems, most of them are: (1) the indiscrimate use of extra-paragraphs, not following a pattern; (2) genera name that are not in italic; (3) small gramatical errors. All the comments are provided in the attached PDF.

There is only one sentence that was not clear to me: Line 233 "Further.....". Please improve it.

I hope my suggestions were useful and contributed to the quality of the study.

Kind regards,

Pedro

Experimental design

no comment

Validity of the findings

no comment

Reviewer 3 ·

Basic reporting

This is a second review. The structure of the article was ameliorated.

Experimental design

Additional details were added where requested by the reviewers. Additional verifications have been made following the suggestions of the reviewers.

Validity of the findings

No comment

Additional comments

Most of the corrections requested by the reviewers were made, but a few new were introduced, including a number of uneeded parentheses around references.
There remains a number of more or less important problems. Is the inclusion of Figure 1 a mistake ? It is redundant with the much better, corrected map of figure 3.
In the additional figures, at least comment on the lack of scale for most of the sample. Is there no way to recover the scale from some detail in your pictures ?
In the introduction, there are double parentheses around several of the references.
L96 replace ; by ,
L130 eliminate this was
L131 first mention of the candidate species, unclear what they are, introduce them briefly. (« species that could not be identified » or something similar, or move the last paragraphe of this parts that introduces them before this phrase)
L133 when not if
L182 primers
L252 galilaeus
L259 hybridization with mitochondrial retention
L281, 287, and elsewhere : uneeded parentheses around references
L295 widely distributed
L215 and elsewhere : clade is not the appropriate term for a distance tree cluster, as already mentionned in the first review (cluster is just fine).
L334 group is needed, why bar it ?
L358 C. gabonensis is dispersed in distinct clusters, not regrouped together. But you cannot deduce polyphyly from a distance tree. It can look like semantics, but it is not, you would need a phylogenetic method for this (parsimony, ML, or bayesian inference).
L405 & 406 Garra

---

## Round 0.3 · accepted · Accept

Dear Authors,

Thank you for implementing the suggestions of the last revision. Your manuscript is now in excellent shape, and I am happy to accept it.

Congratulations on a job well done.

Tomas Hrbek